# 1-(Arylsulfonyl-isoindol-2-yl)piperazines as 5-HT_6_R Antagonists: Mechanochemical Synthesis, In Vitro Pharmacological Properties and Glioprotective Activity

**DOI:** 10.3390/biom13010012

**Published:** 2022-12-21

**Authors:** Vittorio Canale, Wojciech Trybała, Séverine Chaumont-Dubel, Paulina Koczurkiewicz-Adamczyk, Grzegorz Satała, Ophélie Bento, Klaudia Blicharz-Futera, Xavier Bantreil, Elżbieta Pękala, Andrzej J. Bojarski, Frédéric Lamaty, Philippe Marin, Paweł Zajdel

**Affiliations:** 1Department of Organic Chemistry, Faculty of Pharmacy, Jagiellonian University Medical College, 9 Medyczna Street, 30-688 Krakow, Poland; 2Institut de Génomique Fonctionelle, Université de Montpellier, CNRS, INSERM, 34094 Montpellier, France; 3Department of Pharmaceutical Biochemisty, Faculty of Pharmacy, Jagiellonian University Medical College, 9 Medyczna Street, 30-688 Krakow, Poland; 4Department of Medicinal Chemistry, Maj Institute of Pharmacology, Polish Academy of Sciences, 12 Smętna Street, 31-343 Krakow, Poland; 5IBMM, Université de Montpellier, CNRS, ENSCM, 34095 Montpellier, France; 6Institut Universitaire de France (IUF), 75005 Paris, France

**Keywords:** mechanochemical synthesis, 5-HT_6_ receptor inverse agonism/neutral antagonism, Gs and Cdk5 signaling pathways, glioprotective properties, rotenone toxicity

## Abstract

In addition to the canonical Gs adenylyl cyclase pathway, the serotonin type 6 receptor (5-HT_6_R) recruits additional signaling pathways that control cognitive function, brain development, and synaptic plasticity in an agonist-dependent and independent manner. Considering that aberrant constitutive and agonist-induced active states are involved in various pathological mechanisms, the development of biased ligands with different functional profiles at specific 5-HT_6_R-elicited signaling pathways may provide a novel therapeutic perspective in the field of neurodegenerative and psychiatric diseases. Based on the structure of SB-258585, an inverse agonist at 5-HT_6_R-operated Gs and Cdk5 signaling, we designed a series of 1-(arylsulfonyl-isoindol-2-yl)piperazine derivatives and synthesized them using a sustainable mechanochemical method. We identified the safe and metabolically stable biased ligand **3g**, which behaves as a neutral antagonist at the 5-HT_6_R-operated Gs signaling and displays inverse agonist activity at the Cdk5 pathway. Inversion of the sulfonamide bond combined with its incorporation into the isoindoline scaffold switched the functional profile of **3g** at Gs signaling with no impact at the Cdk5 pathway. Compound **3g** reduced the cytotoxicity of 6-OHDA and produced a glioprotective effect against rotenone-induced toxicity in C8-D1A astrocyte cell cultures. In view of these findings, compound **3g** can be considered a promising biased ligand to investigate the role of the 5-HT_6_R-elicited Gs and Cdk5 signaling pathways in neurodegenerative diseases.

## 1. Introduction

The serotonin type 6 receptor (5-HT_6_R) is a Gs-coupled receptor widely expressed in brain regions involved in cognitive functions [1] and has been considered an important target to alleviate cognitive symptoms related to neurodegenerative and psychiatric disorders [2,3,4,5,6]. Several lines of evidence indicate that modulation of cognition by 5-HT_6_R ligands might be independent of its coupling to Gs, suggesting the engagement of alternative signaling mechanisms. Indeed, interactomic studies revealed that the 5-HT_6_R is linked to different GIPs (GPCR interacting proteins) that underlie its modulation of cognitive (mechanistic target of rapamycin (mTOR)) and neuro-developmental (cyclin-dependent kinase 5 (Cdk5) and G protein-regulated inducer of neurite outgrowth 1 (GPR1N1)) processes [7,8]. Recently, a non-canonical 5-HT_6_R-elicited Gαq/11-RhoA pathway, which modulates nuclear actin and increases histone acetylation and chromatin accessibility, has been identified in hippocampal neurons [9]. An important feature of the 5-HT_6_R is its high level of ligand-independent constitutive activity, which corresponds to the ability of a receptor to be active even in the absence of an agonist [10]. The role of 5-HT_6_R constitutive activity at canonical Gs signaling and non-canonical signaling has been described in various pathophysiological conditions [11,12,13,14,15]. Consistent with those findings, the development of 5-HT_6_R ligands acting as inverse agonists and/or neutral antagonists at the different 5-HT_6_R-operated signaling pathways is of utmost interest to decipher the cellular mechanism(s) under the control of the various 5-HT_6_R signal transduction mechanisms and of constitutive vs. agonist-dependent receptor activation.

Many ligands were evaluated to study the role of 5-HT_6_R, yet the design of 5-HT_6_R-biased ligands remains highly challenging. Using molecular dynamic simulations, we previously reported structural hints responsible for inverse agonism and neutral antagonism at Gs signaling in a group of imidazo[4,5-*b*]- and imidazo[4,5-*c*]pyridines (Figure 1A) [16,17]. A different approach, based on the truncation of the planar *1H*-pyrrolo[3,2-*c*]quinoline core present in the 5-HT_6_R antagonist (*S*)-1-[(3-chlorophenyl)sulfonyl]-4-(pyrrolidine-3-yl-amino)-1*H*-pyrrolo[3,2-*c*]quinoline (CPPQ) [18], to the more flexible 2-phenyl-*1H*-pyrrole-3-carboxamide scaffold, shifted the compound activity from neutral antagonism to inverse agonism at receptor-operated Gs and Cdk5 signaling (Figure 1B) [19].

These observations prompted us to investigate further the role of the arylsulfonamide fragment in the development of active 5-HT_6_R ligands. Starting from SB-258585, an inverse agonist at 5-HT_6_R-operated Gs and Cdk5 signaling [20,21], we designed a set of 1-(arylsulfonyl-isoindol-2-yl)piperazines. We focused on the inversion of the sulfonamide bond combined with its incorporation into differently substituted isoindoline scaffold and flipping of the piperazine moiety in the 2-, 3-, or 4- position at the phenyl ring (Figure 2).

The impact of compounds synthesized using a mechanochemical approach on 5-HT_6_R-operated Gs and Cdk5 signaling pathways was examined. The glioprotective properties of the selected compound with the highest potency at Gs and Cdk5 signaling and favorable preliminary ADMET profile were investigated in C8-D1A astrocytes exposed to the gliotoxic treatments using 6-hydroxydopamine (6-OHDA) and rotenone (ROT).

## 2. Material and Methods

### 2.1. Chemistry

#### 2.1.1. General Chemical Methods

All commercially available reagents were of the highest purity from Fluorochem, Across Organic. The milling treatments were carried out in a vibratory ball-mill Retsch MM400 operated at 30 Hz. The milling load was defined as the sum of the mass of the reactants per free volume in the jar and was equal to 10 mg/mL. All reactions using the vibratory ball mill were performed under air.

Mass spectra were recorded on a UPLCMS/MS system consisting of a Waters ACQUITY UPLC (Waters Corporation, Milford, MA, USA) coupled to a Waters TQD mass spectrometer (electrospray ionization mode ESI-tandem quadrupole). Chromatographic separations were carried out using the Acquity UPLC BEH (bridged ethyl hybrid) C18 column; 2.1 mm × 100 mm, and 1.7 μm particle size, equipped with Acquity UPLC BEH C18 Van Guard precolumn; 2.1 mm × 5 mm, and 1.7 μm particle size. The column was maintained at 40 °C and eluted under gradient conditions from 95% to 0% of eluent A over 10 min, at a flow rate of 0.3 mL min^−1^. Eluent A: water/formic acid (0.1%, *v*/*v*), Eluent B: acetonitrile/formic acid (0.1%, *v*/*v*).

^1^H and ^13^C NMR spectra were recorded on a JEOL JNM-ECZR500 RS1 (ECZR version) at 500 and 126 MHz, respectively, and were reported in ppm using deuterated solvent for calibration (CDCl_3_). The *J* values were reported in hertz (Hz), and the splitting patterns were designated as follows: br s. (broad singlet), s (singlet), d (doublet), t (triplet), q (quartet), dd (doublet of doublets), ddd (doublet of doublet of doublets), m (multiplet).

#### 2.1.2. General Procedure A for Sulfonylation of Isoindoline Derivatives to Obtain Intermediates **2a**–**l**

Intermediates **1a**–**d** (1 eq), different fluoro-substituted benzenesulfonyl chloride (1.1 eq), and NaOH (3 eq) were introduced in a 35 mL SS jar (milling load 10 mg/mL) with one SS ball (ϕ_ball_ = 1.5 cm). The reaction was carried out for 5–7 min at rt. The mixture was transferred into a filtration funnel, washed with distilled water, and dried under reduced pressure yielding intermediate **2a**–**l** in high yields (85–98%).

#### 2.1.3. General Procedure B for Aromatic Substitution to Obtain Final Compounds **3a**–**l**

Intermediates **2a**–**l** (1 eq) and anhydrous piperazine (3 eq) were introduced in a 10 mL SS jar (milling load 10 mg/mL) with one stainless steel ball (ϕ_ball_ = 1.5 cm) followed by the addition of MeCN (34 μL, η = 0.4 μL mg^−1^) as a liquid assistant. The reaction was carried out for 1.5 h at 50 °C. In the case of intermediates **2e**, **2f**, **2g**, and **2h**, DMSO (42 μL, η = 0.4 μL mg^−1^) were used as a liquid additive, and the reactions were milled for 1.5 h at 80 °C. After the addition of cold water (5 mL), the resulting mixture was transferred into a filtration funnel, washed with distilled water, and dried under reduced pressure. The products were purified via crystallization in ethanol, yielding final compounds **3a**–**l** as a white powder (yields 75–88%).

#### 2.1.4. General Procedure C for One-Pot Two-Step Reaction for Obtaining Compounds **3e**, **3f**, and **3g**

Isoindoline derivative (1 eq), sodium hydroxide (3 eq), and 3-fluorobenzenesulfonyl chloride (1.1 eq) were introduced in a 35 mL SS jar (milling load 10 mg/mL) with one stainless steel ball (ϕ_ball_ = 1.5 cm). After 10 min of milling at rt, anhydrous piperazine (3 eq) and DMSO (140 μL, η = 0.4 μL mg^−1^) were added into the jar, and the reaction was carried out for an additional 60 min at 80 °C. The resulting mixture was transferred into a filtration funnel, washed with distilled water, and dried under reduced pressure. The products were purified via crystallization in ethanol, yielding final compounds **3e**, **3f**, and **3g** as white powders (yields 85–88%).

#### 2.1.5. Characterization Data for Final Compounds

##### 2-{[4-(Piperazin-1-yl)phenyl]sulfonyl}isoindoline **3a**

White solid, 34 mg (isolated yield 88%) following procedure B; UPLC/MS purity 100%, *t*_R_ = 4.43; C_18_H_21_N_3_O_2_S, MW 343.45, Monoisotopic Mass 343.14, [M+H]^+^ 344.2. ^1^H NMR (500 MHz, CDCl_3_) δ ppm 1.70 (br s, 1H), 2.95–3.00 (m, 4H), 3.22–3.30 (m, 4H), 4.57 (s, 4H), 6.85–6.90 (m, 2H), 7.12–7.16 (m, 2H), 7.18–7.23 (m, 2H), 7.67–7.76 (m, 2H). ^13^C NMR (126 MHz, DMSO-*d*_6_) δ ppm 45.8, 48.0, 53.9, 113.9, 123.2, 123.3, 128.1, 129.7, 136.5, 154.5.

##### 4-Chloro-2-{[4-(piperazin-1-yl)phenyl]sulfonyl}isoindoline **3b**

White solid, 33 mg (isolated yield 81%) following procedure B; UPLC/MS purity 100%, *t*_R_ = 4.97; C_18_H_20_ClN_3_O_2_S, MW 377.89, Monoisotopic Mass 377.10, [M+H]^+^ 378.2. ^1^H NMR (500 MHz, DMSO-*d*_6_) δ ppm 2.73 (br s, 4H), 3.15 (br s, 4H), 4.45 (s, 2H), 4.55 (s, 2H), 6.96 (d, *J* = 8.9 Hz, 2H), 7.16–7.29 (m, 3H), 7.61 (d, *J* = 8.9 Hz, 2H). ^13^C NMR (126 MHz, DMSO-*d*_6_) δ ppm 45.8, 48.0, 53.4, 54.7, 113.9, 122.3, 122.9, 128.0, 128.2, 129.7, 130.4, 134.8, 139.1, 154.6.

##### 4-Bromo-2-{[4-(piperazin-1-yl)phenyl]sulfonyl}isoindoline **3c**

White solid 35 mg (isolated yield 80%) following procedure B; UPLC/MS purity 99%, *t*_R_ = 5.35; C_18_H_20_BrN_3_O_2_S, MW 422.34, Monoisotopic Mass 421.05, [M+H]^+^ 422.0/424.0. ^1^H NMR (500 MHz, DMSO-*d*_6_) δ ppm 2.68–2.80 (m, 4H), 3.10–3.21 (m, 4H), 4.40 (s, 2H), 4.58 (s, 2H), 6.97 (d, *J* = 9.2 Hz, 2H), 7.16 (t, *J* = 8.6 Hz, 1H), 7.22 (d, *J* = 9.2 Hz, 1H), 7.39 (d, *J* = 7.7 Hz, 1H), 7.61 (d, *J* = 8.9 Hz, 2H). ^13^C NMR (126 MHz, DMSO-*d*_6_) δ ppm 45.8, 48.0, 54.9, 55.1, 113.9, 116.9, 122.8, 122.9, 129.7, 130.6, 130.9, 136.9, 138.8, 154.6.

##### 5-Bromo-2-{[4-(piperazin-1-yl)phenyl]sulfonyl}isoindoline **3d**

White solid, 36 mg (isolated yield 82%) following procedure B; UPLC/MS purity 100%, *t*_R_ = 5.18; C_18_H_20_BrN_3_O_2_S, MW 422.34, Monoisotopic Mass 421.05, [M+H]^+^ 422.1/424.1. ^1^H NMR (500 MHz, DMSO-*d*_6_) δ ppm 2.73 (br s, 4H), 3.15 (br s, 4H), 4.41 (s, 2H), 4.45 (s, 2H), 6.96 (d, *J* = 9.2 Hz, 2H), 7.16 (d, *J* = 8.0 Hz, 1H), 7.37 (d, *J* = 8.3 Hz, 1H), 7.43 (s, 1H), 7.57 (d, *J* = 8.0 Hz, 2H). ^13^C NMR (126 MHz, DMSO-*d*_6_) δ ppm 45.8, 48.0, 53.6, 113.9, 120.9, 123.0, 125.5, 126.4, 129.7, 130.9, 136.1, 139.4, 154.5.

##### 2-{[3-(Piperazin-1-yl)phenyl]sulfonyl}isoindoline **3e**

White solid, 137 mg (isolated yield 85%) following procedure C; UPLC/MS purity 100%, *t*_R_ = 4.52; C_18_H_21_N_3_O_2_S, MW 343.45, Monoisotopic Mass 343.14, [M+H]^+^ 344.2. ^1^H NMR (500 MHz, CDCl_3_) δ ppm 2.10 (br. s, 1H), 3.00–3.05 (m, 4H), 3.15–3.22 (m, 4H), 4.62 (s, 4H), 7.05 (ddd, *J* = 8.1, 2.5, 0.9 Hz, 1H), 7.14–7.18 (m, 2H), 7.20–7.23 (m, 2H), 7.28–7.31 (m, 1H), 7.33–7.37 (m, 2H). ^13^C NMR (126 MHz, CDCl_3_) δ ppm 45.9, 49.6, 53.8, 114.0, 117.9, 119.7, 122.7, 127.8, 130.0, 136.2, 137.3, 152.2.

##### 4-Chloro-2-{[3-(piperazin-1-yl)phenyl]sulfonyl}isoindoline **3f**

White solid, 145 mg (isolated yield 86%) following procedure C; UPLC/MS purity 99%, *t*_R_ = 5.08; C_18_H_20_ClN_3_O_2_S, MW 377.89, Monoisotopic Mass 377.10, [M+H]^+^ 378.2. ^1^H NMR (500 MHz, CDCl_3_) δ ppm 2.12 (br s, 1H), 2.99–3.08 (m, 4H), 3.15–3.25 (m, 4H), 4.63 (s, 2H), 4.67 (s, 2H), 7.01–7.09 (m, 2H), 7.14–7.21 (m, 2H), 7.27–7.31 (m, 1H), 7.32–7.40 (m, 2H). ^13^C NMR (126 MHz, CDCl_3_) δ ppm 45.9, 49.5, 53.5, 54.5, 113.9, 117.8, 119.8, 120.9, 127.8, 129.1, 129.5, 130.1, 135.1, 137.3, 138.1, 152.2.

##### 4-Bromo-2-{[3-(piperazin-1-yl)phenyl]sulfonyl}isoindoline **3g**

White solid, 133 mg (isolated yield 88%) following procedure C; UPLC/MS purity 100%, *t*_R_ = 5.17; C_18_H_20_BrN_3_O_2_S, MW 422.34, Monoisotopic Mass 421.05, [M+H]^+^ 422.0/424.0. ^1^H NMR (500 MHz, DMSO-*d*_6_) δ ppm 2.77 (br s, 4H), 3.04 (br s, 4H), 4.48 (s, 2H), 4.66 (s, 2H), 7.16 (t, *J* = 5.7 Hz, 4H), 7.22 (d, *J* = 8.0 Hz, 2H), 7.33–7.45 (m, 2H). ^13^C NMR (126 MHz, DMSO-*d*_6_) δ ppm 45.9, 49.2, 55.0, 55.2, 112.7, 116.9, 117.2, 120.0, 122.8, 130.6, 130.7, 130.9, 136.8, 136.9, 138.8, 152.5.

##### 5-Bromo-2-{[3-(piperazin-1-yl)phenyl]sulfonyl}isoindoline **3h**

White solid, 33 mg (isolated yield 76%) following procedure B; UPLC/MS purity 100%, *t*_R_ = 5.23; C_18_H_20_BrN_3_O_2_S, MW 422.34, Monoisotopic Mass 421.05, [M+H]^+^ 422.1/424.1. ^1^H NMR (500 MHz, DMSO-*d*_6_) δ ppm 2.77 (br s, 4H), 3.04 (br s, 4H), 4.47 (s, 2H), 4.52 (s, 2H), 7.10–7.25 (m, 4H), 7.33–7.41 (m, 2H), 7.43 (s, 1H). ^13^C NMR (126 MHz, DMSO-*d*_6_) δ ppm 45.9, 49.1, 53.6, 53.7, 112.8, 117.2, 120.0, 121.0, 125.5, 126.4, 130.6, 130.9, 136.0, 136.9, 139.3, 152.5.

##### 2-{[2-(Piperazin-1-yl)phenyl]sulfonyl}isoindoline **3i**

White solid, 33 mg (isolated yield 84%) following procedure B; UPLC/MS purity 100%, *t*_R_ = 4.59; C_18_H_21_N_3_O_2_S, MW 343.45, Monoisotopic Mass 343.14, [M+H]^+^ 344.3. ^1^H NMR (500 MHz, CDCl_3_) δ ppm 1.91 (br.s, 1H), 2.90 (t, *J* = 4.3 Hz, 4H), 3.03 (t, *J* = 4.6 Hz, 4H), 4.82 (s, 4H), 7.13–7.16 (m, 2H), 7.19–7.23 (m, 3H), 7.28–7.31 (m, 1H), 7.44–7.53 (m, 1H), 8.02 (dd, *J* = 8.0, 1.7 Hz, 1H). ^13^C NMR (126 MHz, CDCl_3_) δ ppm 46.1, 54.1, 55.3, 122.6, 123.8, 124.8, 127.6, 132.2, 134.0, 134.4, 136.6, 153.2.

##### 4-Chloro-2-{[2-(piperazin-1-yl)phenyl]sulfonyl}isoindoline **3j**

White solid, 37 mg (isolated yield 83%) following procedure B; UPLC/MS purity 99%, *t*_R_ = 5.09; C_18_H_20_ClN_3_O_2_S, MW 377.89, Monoisotopic Mass 377.10, [M+H]^+^ 378.2. ^1^H NMR (500 MHz, CDCl_3_) δ ppm 2.55–2.62 (m, 1H), 2.93–3.00 (m, 4H), 3.09 (t, *J* = 4.3 Hz, 4H), 4.80–4.85 (m, 2H), 4.85–4.88 (m, 2H), 6.97–7.10 (m, 1H), 7.14–7.20 (m, 2H), 7.21–7.25 (m, 1H), 7.32 (dd, *J* = 8.0, 1.1 Hz, 1H), 7.48–7.54 (m, 1H), 8.02 (dd, *J* = 8.0, 1.7 Hz, 1H). ^13^C NMR (126 MHz, CDCl_3_) δ ppm 45.9, 53.7, 54.7, 55.0, 120.8, 123.9, 125.0, 127.7, 129.0, 129.4, 132.2, 134.1, 134.2, 135.4, 138.4, 152.9.

##### 4-Bromo-2-{[2-(piperazin-1-yl)phenyl]sulfonyl}isoindoline **3k**

White solid, 38 mg (isolated yield 88%) following procedure B; UPLC/MS purity 100%, *t*_R_ = 5.25; C_18_H_20_BrN_3_O_2_S, MW 422.34, Monoisotopic Mass 421.05, [M+H]^+^ 422.1/424.1. ^1^H NMR (500 MHz, DMSO-*d*_6_) δ ppm 2.77 (t, *J* = 4.6 Hz, 4H), 3.19 (t, *J* = 4.9 Hz, 4H), 4.40 (s, 2H), 4.58 (s, 2H), 6.98 (d, *J* = 9.2 Hz, 2H), 7.17 (t, *J* = 7.7 Hz, 1H), 7.22 (d, *J* = 7.2 Hz, 1H), 7.40 (d, *J* = 8.6 Hz, 1H), 7.59–7.64 (m, 2H). ^13^C NMR (126 MHz, DMSO-*d*_6_) δ ppm 45.5, 47.6, 54.9, 55.1, 114.0, 116.9, 122.8, 123.1, 129.7, 130.6, 130.9, 136.9, 138.8, 154.4.

##### 5-Bromo-2-{[2-(piperazin-1-yl)phenyl]sulfonyl}isoindoline **3l**

White solid, 37 mg (isolated yield 86%) following procedure B; UPLC/MS purity 100%, *t*_R_ = 5.28; C_18_H_20_BrN_3_O_2_S, MW 422.34, Monoisotopic Mass 421.05, [M+H]^+^ 422.1/424.1. ^1^H NMR (500 MHz, DMSO-*d*_6_) δ ppm 2.68 (t, *J* = 4.3 Hz, 4H), 2.79 (t, *J* = 4.3 Hz, 4H), 4.68 (s, 4H),7.18 (d, *J* = 8.0 Hz, 1H), 7.27 (t, *J* = 7.6 Hz, 1H), 7.34–7.40 (m, 2H), 7.41–7.48 (m, 1H), 7.52–7.61 (m, 1H), 7.87 (dd, *J* = 8.0, 1.4 Hz, 1H). ^13^C NMR (126 MHz, DMSO-*d*_6_) δ ppm 45.9, 53.7, 53.8, 55.5, 120.9, 124.7, 125.3, 125.4, 126.2, 130.8, 132.4, 133.8, 135.0, 136.4, 139.6, 153.6.

### 2.2. In Vitro Biological Evaluation

#### 2.2.1. Radioligand Binding Assays

All experiments were performed in HEK-293 cells, which stably express the human 5-HT_1A_, 5-HT_6_, 5-HT_7b_, and D_2L_ receptors, or using CHO-K1 cells expressing human serotonin 5-HT_2A_ receptor, following the previously reported procedures [22,23,24]. For displacement studies, the following radioligands (PerkinElmer, USA) at given concentrations were used at: 2.5 nM [^3^H]-8-OH-DPAT (135.2 Ci/ mmol); 1 nM [^3^H]-ketanserin (53.4 Ci/mmol); 2 nM [^3^H]-LSD (83.6 Ci/mmol); 0.8 nM [^3^H]-5-CT (39.2 Ci/mmol) or 2.5 nM [^3^H]-raclopride (76.0 Ci/mmol) for 5-HT_1A_, 5-HT_2A_, 5-HT_6_, 5-HT_7b_ and D_2L_ receptors, respectively. All tested compounds were evaluated in triplicate at 7 concentrations (10^−10^–10^−4^ M). The inhibition constants (*K*_i_) were calculated from the Cheng–Prusoff equation [25]. A detailed description is reported in the Appendix A.

#### 2.2.2. Impact of Evaluated Compounds on cAMP Production in 1321N1 Cells

The ability of compounds **3e**, **3f**, and **3g** to inhibit 5-CT-induced production of cAMP was assessed using 1321N1 cells expressing the human 5-HT_6_R (PerkinElmer) using previously described procedures [15,17]. The level of cAMP was measured using the LANCE cAMP detection kit (PerkinElmer) according to the manufacturer’s protocol. TR-FRET (Time Resolved Fluorescence Resonance Energy Transfer) was detected by an Infinite M1000 Pro (Tecan) using instrument settings from the LANCE cAMP detection kit manual. Tested compounds were evaluated at 8 concentrations (10^−11^–10^−4^ M) in triplicate. *K*_b_ values were calculated from the Cheng–Prusoff equation [25]. A detailed description is reported in the Appendix A.

#### 2.2.3. Impact of Evaluated Compounds on cAMP Production Elicited by Constitutively Active 5-HT_6_R in NG108-15 Cells

The functional properties at Gs signaling of **3e**, **3f**, **3g**, and the prototypic inverse agonist SB-258585 were evaluated using NG108-15 cells transiently expressing 5-HT_6_R and the cAMP sensor CAMYEL (cAMP sensor using YFP-Epac-RLuc) [26]. We previously demonstrated that the 5-HT_6_R displays a high level of constitutive activity in this model. Changes in cyclic AMP levels upon exposure to increasing concentrations of the tested compounds were assessed by BRET (Bioluminescence Resonance Energy Transfer) measurement using a Mithras LB 940 plate reader (Berthold Technologies), as previously described [17]. A detailed description is reported in the Appendix A.

#### 2.2.4. Impact of Tested Compounds on Cdk5-Dependent Neurite Growth

The inverse agonist properties at Cdk5 signaling of **3e**, **3f**, **3g**, and the prototypic inverse agonist SB-258585 were evaluated in NG108-15 cells transiently expressing 5-HT_6_R by measuring neurite outgrowth, as previously reported. NG108-15 cells were transfected with plasmids encoding either cytosolic GFP or a GFP-tagged 5-HT_6_R and grown on glass coverslips for 6 h. Then, cells were treated with either vehicle or the evaluated compounds. Neurite growth was measured on cells imaged using an AxioImagerZ1 microscope equipped with epifluorescence (Zeiss) using the Neuron J plugin of the ImageJ software (NIH). A detailed description is reported in the Appendix A.

#### 2.2.5. In Vitro Assessment of Metabolic Stability

Metabolic stability studies were performed according to reported protocols [27,28]. Evaluated compounds **3e**, **3f**, and **3g** at the final concentration of 20 µM were preincubated in phosphate buffer (pH = 7.4), containing rat liver microsomes (microsome from rat male liver, pooled; 0.5 mg/mL; Merck/Millipore Sigma, Darmstadt, Germany, St. for 10 min at 37 °C. The reaction was initiated by adding the NADPH-regenerating system. After 0, 30, and 60 min, reactions were quenched with ice-cold methanol containing internal standard (pentoxifylline @ 100 nM). Next, samples were centrifuged, and the supernatants were analyzed by UPLC/MS. T_1/2_ was determined from the slope of the line on Ln (%remaining of parent compounds) vs. time plots. Cl_int_ was calculated from the equation: Cl_int_ = [volume of incubation (μL)/amount of protein (mg) × 0.693]/t_1/2_. All samples were analyzed in duplicate. The assay performance was confirmed by using extensive or low metabolized drugs as references (Imipramine and Donepezil, respectively).

#### 2.2.6. In Vitro Cytotoxicity Evaluation

Hepatotoxicity, neurotoxicity, and glial cytotoxicity were investigated using the following cellular models: human hepatocellular carcinoma (HepG2), human neuroblastoma (SHSY-5Y), and mouse astrocytes (C8-D1A), all from ATCC (American Type Culture Collection, Manassas, VA, USA). Cells were cultured in appropriate culture media recommended by ATCC supplemented with 10% fetal bovine serum (FBS; Gibco, Life Technologies, Carlsbad, CA, USA) and 1% antibiotic mixture (Gibco, Life Technologies, Carlsbad, CA, USA). Cells were cultured at standard conditions (5% CO_2_, 95% humidity, 37 °C). For cytotoxicity experiments, cells were seeded into 96 well plates at an initial density of 2000 cells/well. After overnight incubation, cells were treated with **3e**, **3f**, and **3g** in a concentration range of 0.78–50 µM for 48 h; then, an MTT assay was performed as described previously [17,29]. Briefly, following cell exposure, 10 µL of MTT reagent was added to each well. After the next 4 h, the medium was aspirated, and formazan produced in cells appeared as dark crystals in the bottom of the wells. Next, 100 µL DMSO was added to each well. Then, the optical density of the solution (OD) at 570 nm was determined on a plate reader (Spectra Max iD3, Molecular Devices, San Jose, CA, USA). The experiments were run three times in triplicates.

#### 2.2.7. In Vitro Assessment of Glioprotective Properties

Mouse astrocytes (C8-D1A cell line, ATCC) were cultured in DMEM (Dulbecco’s Modified Eagle’s Medium) supplemented with 10% FBS (FBS; Gibco, Life Technologies, Carlsbad, CA, USA) and 1% antibiotic mixture (Gibco, Life Technologies, Carlsbad, CA, USA) at standard condition until the cells reached 80–90% confluence. Cells were harvested and seeded in 96-well plates at a density of 5000 cells/well. All experiments were conducted with a reduced amount of serum equal to 2.5%. The optimal amount of serum was determined experimentally on the basis of preliminary studies. After 24 h, cells were pre-incubated in the presence of compound **3g** or references CPPQ, SB-258585, and WAY-181187 (all at 0.25 µM). Then 6-OHDA (20 µM) or rotenone (0.5 µM) were added and co-incubated for the next 24 h. The ability of tested compounds to prevent 6-OHDA-induced or rotenone-induced cytotoxicity was assessed by MTT assay following the protocol described above. The experiments were run three times in triplicates.

## 3. Result and Discussion

### 3.1. Mechanochemical Synthesis of Compounds **3a**–**3l**

A medicinal mechanochemical approach [28,30,31,32] was employed for the synthesis of a novel group of 1-(arylsulfonyl-isoindol-2-yl)piperazines **3a**–**3l** (Figure 1). The optimization of the synthetic process started with the sulfonylation of unsubstituted isoindoline with the 4-fluorobenzenesulfonyl chloride (1.1 eq) (Appendix A). The reaction was initially performed in a 10 mL stainless-steel (SS) jar with a 1.5 cm diameter ball of the same material by using a vibratory ball mill (vbm) operated at 30 Hz. Potassium carbonate and sodium hydroxide guaranteed high conversions of reagents with a milling time of only 5 min. Although the use of potassium carbonate provided a better conversion rate, sodium hydroxide was chosen for further optimization due to its lower molecular weight. Increasing the amount of base from 2 to 3 equivalents resulted in full conversion to intermediate **2a** (isolated yield 98%). The reaction was then scaled up using a 35 mL SS jar without affecting the conversion or the yield (Appendix A). The optimized reaction conditions for sulfonylation on a larger scale were then applied for the generation of intermediates **2b**–**2l**. Regardless of the substituent at the isoindoline scaffold and the kind of arylsulfonyl chloride, all intermediates **2a**–**2l** were obtained after a short milling time, ranging from 5 to 7 min. Of note, products **2a**–**2l** were isolated after filtration of the inorganic base, followed by washing the filtrate with water to provide desired compounds in high yields (85–98%).

Next, a mechanochemical procedure was applied to perform a nucleophilic aromatic substitution (S_N_Ar) between fluorinated derivatives **2a**–**2l** and anhydrous piperazine to obtain final compounds **3a**–**3l**. The advantages of using temperature-controlled mechanochemical procedures have been recently demonstrated for different organic reactions [33,34,35,36]. In line with those findings, heating of a 10 mL SS jar containing intermediate **2a** (1 eq) and anhydrous piperazine (1 eq) at 50 °C for 1.5 h by using a temperature-controlled heat gun pre-set at 90 °C enabled the formation of the product **3a** (55% of conversion). No product was observed after milling of reagents at room temperature (Appendix A). Because the conversion rate was still not satisfactory, MeCN, a widely used aprotic polar solvent for S_N_Ar reactions in solution, was evaluated as a non-toxic liquid additive to enhance the overall mixing and reaction kinetics. Employing liquid-assisted grinding (LAG), in tandem with the increase in the amount of amine (3 eq of anhydrous piperazine), drastically improved the conversion rates up to 90% to furnish the desired product **3a** in 85% yield (Appendix A). Regardless of the kind of substituent at the isoindoline core, similar results were obtained with different 4-fluoro and 2-fluoro-containing substrates (isolated yields 80–88%). Despite the low reactivity of the fluorine atom in the 3-position with respect to sulfonamide moiety, optimization of reaction conditions was required. The replacement of MeCN with DMSO as a liquid additive enabled to increase the milling temperature of the jar to 80 °C (heat gun pre-set at 120 °C), providing final compounds **3e**, **3f**, **3g**, and **3h** in acceptable yields (75–78%). All the final compounds were isolated after simple precipitation from water followed by filtration to remove the excess of piperazine and crystallization in ethanol (Appendix A).

### 3.2. Radioligand Binding Assays in HEK-293 Cells and SAR Studies

All synthesized compounds were evaluated in ^3^[H]-LSD competition binding experiments for their affinity for 5-HT_6_R expressed in HEK-293 cells (Table 1). Embedding the sulfonamide moiety into an unsubstituted isoindoline scaffold was sufficient to provide potent compounds **3a** and **3e**. Compounds bearing the chlorine or bromine atom at the 4- or 5-position of isoindoline scaffold were equipotent (**3b** and **3c** vs. **3a**) or showed lower affinity than their unsubstituted analogs (**3d** vs. **3a** and **3l** vs. **3j**). Regardless of the kind of halogen at the isoindoline core, 3-piperazinyl derivatives **3e**, **3f**, **3g**, and **3h** were the most potent compounds (*K*_i_ < 5 nM). A shift of the piperazine moiety from 3- to 4-position at the phenyl ring slightly decreased the affinity for the 5-HT_6_R but still provided high-affinity ligands **3a**, **3b**, and **3c**. In contrast, the introduction of piperazine moiety at the 2-position at the phenyl ring was not beneficial for the interaction with the 5-HT_6_R, as compounds **3i**, **3j** and **3l** were not active (*K*_i_ > 900 nM). An exception, compound **3k** bearing the 4-bromo substituent at the isoindoline scaffold, displayed average affinity at the 5-HT_6_R. This effect might result from the favorable localization of the bromine atom at the isoindoline fragment, which enabled compound **3k** to adopt an optimal position in the receptor binding pocket via the formation of hydrophobic/steric interactions.

The most potent 5-HT_6_R ligands **3e**, **3f**, and **3g** bearing the piperazine moiety localized at position-3 at the sulfonamide fragment were then tested for their activity at serotonin 5-HT_1A_, 5-HT_2A_, 5-HT_7_, and dopaminergic D_2_ receptors (Table 2). These compounds were highly selective over the off-target 5-HT_7_R and D_2_R (*K*_i_ > 2 µM). They also displayed good selectivity toward the 5-HT_1A_R subtype (up to 680-folds). The selectivity index over the 5-HT_2A_R (*K*_i_ < 200 nM) was still satisfactory.

### 3.3. One-Pot Two-Step Mechanochemical Synthesis of Compounds **3e**, **3f**, and **3g**

To optimize the developed mechanochemical procedure, a one-pot, two-step protocol was employed for the synthesis of the most potent derivatives **3e**, **3f**, and **3g** (Figure 2).

According to the newly elaborated protocol, intermediates generated upon milling the isoindolines **1a**–**c** (1 eq.) with 3-fluorobenzenesulfonyl chloride (1.1 eq) and NaOH (3 eq) were directly submitted to S_N_Ar by the addition of anhydrous piperazine (3 eq) and DMSO (η = 0.4 µL/mg) in the 35 mL SS jar (Figure 2). The simplified one-pot two-step synthetic procedure enabled to obtain final compounds in higher overall yields (85–88%) when compared to the sequential two-step protocol (75–78%) by improving the conversion rates of the S_N_Ar reaction. A possible explanation could be found in the role of NaOH in neutralizing the generated in situ HF, maintaining a favorable basic environment for the substitution reaction.

### 3.4. Antagonist Properties of Selected Compounds at 5-HT_6_R-Operated Gs Signaling

The antagonist properties of compounds **3e**, **3f**, and **3g** at 5-HT_6_R-operated Gs signaling were assessed in comparison with the reference ligand intepirdine and using 5-CT as an agonist in 1321N1 cells over-expressing the 5-HT_6_R (Figure 3A). Likewise, intepirdine (*K*_b_ = 1 ± 0.4 nM, Figure 3A), all compounds inhibited the 5-CT-stimulated cAMP accumulation and, thus, were classified as receptor antagonists (*K*_b_ = 0.7−6.9 nM, Table 2).

Subsequently, the ability of compounds **3e**, **3f**, and **3g** to inhibit 5-HT_6_R constitutive activity were further investigated in neuroblastoma NG108-15 cells, in line with our previous observation indicating that transiently expressed receptors exhibit a high level of constitutive activity at Gs signaling in this cell population. Neither of the evaluated compounds significantly affected basal cAMP production in NG108-15 cells, indicating that they behave as neutral antagonists (Table 2, Figure 3B). In contrast and consistent with previous findings, the reference compound SB-258585 reduced the basal level of cAMP, showing inverse agonist properties [14].

In light of these findings and considering structure-functional activity relationships, it seems that the more flexible primary sulfonamide of SB-258585 provides a particular orientation of the aromatic fragment that allows inhibition of the spontaneously active conformation of 5-HT_6_R [20], while inversion of the sulfonamide and its concomitant incorporation into isoindoline scaffold confers neutral antagonist properties.

Beyond the ability to adopt different conformational states able to recruit the canonical Gs-adenylyl cyclase pathway in the presence and absence of an agonist, 5-HT_6_R also activates Cdk5 signaling in an agonist-independent manner [21]. We previously showed that neurite growth in NG108-15 cells transiently expressing the 5-HT_6_R provides an in vitro model to assess agonist-independent activation of Cdk5 signaling, enabling pharmacological distinction between inverse agonists and neutral antagonists at Cdk5 signaling [21,37]. 

As previously observed, transient expression of the 5-HT_6_R in NG108-15 cells markedly increased neurite growth compared with cells expressing GFP alone. Treatment of NG108-15 cells expressing the receptor with compounds **3e**, **3f**, and **3g** for 24 h significantly reduced neurite length (11.81 ± 2.61 μm, n = 112 neurites, 10.92 ± 0.89 μm n = 182 neurites and 8.08 ± 0.84 μm, n = 105 neurites*,* respectively vs. 16.95 ± 1.5 μm, n = 228 neurites in vehicle-treated cells, Figure 4). Of note, compound **3g** was the most effective compound providing a 50% reduction in neurite growth slightly higher than that measured in cells treated with the reference SB-258585 (10.3± 0.77 μm, n = 182 neurites). In this case, reducing the flexibility of the sulfonamide fragment of SB-258585 by its inversion and incorporation into isoindoline moiety did not impact the functional profile of tested compounds at the 5-HT_6_R-elicited Cdk5 pathway. To our knowledge, compound **3g** represents a unique example of a 5-HT_6_R ligand that behaves as a neutral antagonist at Gs signaling and displays potent inverse agonist properties at a non-canonical pathway engaged by the receptor. 

### 3.5. In Vitro Metabolic Stability and Preliminary Safety Assessment for **3e**, **3f**, and **3g**

Preliminary ADME/Tox properties of compounds **3e**, **3f**, and **3g** were screened using in vitro methods. First, biotransformation studies using rat liver microsomes (RLM) revealed that all tested compounds were metabolically stable (Table 3). Unsubstituted derivative **3e** displayed a lower clearance value (Cl_int_ = 3.78 µL/min/mg) than its halogenated analogs **3f** (4-Cl) and **3g** (4-Br), similar to that of the reference drug donepezil (Cl_int_ = 3.65 µL/min/mg). Imipramine, an extensively metabolized drug used as a positive control, displayed much higher clearance (Cl_int_ = 115.65 µL/min/mg).

Then, to exclude potential cytotoxic effects, compounds **3e**, **3f**, and **3g** were tested in human hepatocellular carcinoma (HepG2), human neuroblastoma (SH-SY5Y), and mouse astrocytes (C8-D1A). None of the evaluated compounds affected the metabolic activity of cells, as assessed by the MTT (3-[4,5-dimethylthiazol-2-yl]-2,5-diphenyl tetrazolium bromide) test. None of them induced any hepatotoxic, neurotoxic, or gliotoxic activity at a concentration of 25 µM (Table 3), in contrast to doxorubicin used as a positive control, which displayed high cytotoxic effects (IC_50_ = 4.4, 10.8 and 12.3 µM against SH-SY5Y, HepG2 and C8-D1A cells, respectively).

### 3.6. Glioprotective Properties of Compound **3g**

Considering the pivotal physiological role of astrocytes in contrasting oxidative stress and promoting tissue repair [38,39], supporting their neuroprotective function might be regarded as a tangible strategy to improve the current therapeutic needs in the treatment of neurodegenerative disorders. We next investigated the ability of compound **3g** to protect astrocytes against 6-OHDA- and ROT-induced cytotoxicity in C8-D1A cells, in line with our previous findings indicating that blocking of 5-HT_6_R results in glioprotective effect against toxic insults [17,40]. In comparison, we assessed the effect of WAY-181187, a 5-HT_6_R agonist, SB-258585, an inverse agonist, and CPPQ, a neutral antagonist [18]. For this purpose, cells were co-incubated with the evaluated compounds (at the non-toxic concentration of 0.25 µM) and 6-OHDA or ROT (20 and 0.5 µM, respectively). After 24 h of incubation, the viability of astrocytes was determined using the MTT assay.

Both compound **3g** and CPPQ, which behave as neutral antagonists at the 5-HT_6_R-operated Gs signaling pathway, decreased the cytotoxicity of 6-OHDA and ROT in C8-D1A astrocytes (Figure 5A,B), whereas the inverse agonist SB-258585 did not evoke any protective effect. Likewise, treatment of C8-D1A cells with WAY-181187, a 5-HT_6_R agonist, slightly but not significantly enhanced the gliotoxic effects of 6-OHDA and ROT. These findings suggest that 5-HT_6_R activation by 5-HT contained in the serum from the cell-growing medium contributes to the toxicity of 6-OHDA and rotenone in C8-D1A astrocytes.

## 4. Conclusions

Structural modifications around the sulfonamide moiety of SB-258585 led to the design of 1-(arylsulfonyl-isoindol-2-yl)piperazines as a new framework for developing biased 5-HT_6_R ligands. An application of a one-pot two-step mechanochemical procedure enabled the synthesis of selected derivatives in a fast and efficient manner (isolated yields 85–88%). Tested compounds displayed high metabolic stability in RLM assays and no toxic effect toward human hepatocellular carcinoma and neuroblastoma or a mouse astrocyte cell line. In vitro functional evaluation identified compounds that behave as potent neutral antagonists at 5-HT_6_R-elicited Gs signaling and simultaneously display inverse agonist properties at the Cdk5 pathway. These findings underline the important role of the sulfonamide moiety, in particular its inversion and rigidification, for the design of biased ligands at 5-HT_6_R-operated Gs and Cdk5 signaling pathways. Moreover, the neutral antagonists **3g** and CPPQ exerted glioprotective properties against 6-OHDA-and ROT-induced toxicity in the C8-D1A astrocyte cell line. In contrast, no protective effect and even a slight increase in cytotoxicity were observed upon treatment with the inverse agonist SB-258585 or the agonist WAY-181187, respectively. The identified biased ligand **3g** might be considered as a molecular probe to explore the role of 5-HT_6_R in neurodegenerative diseases further.

## Data Availability

The data presented in this study are available in the Appendix A.

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
