# Peer review of "1-(Arylsulfonyl-isoindol-2-yl)piperazines as 5-HT6R Antagonists: Mechanochemical Synthesis, In Vitro Pharmacological Properties and Glioprotective Activity"

_biomolecules, 2022, doi:10.3390/biom13010012_

Round 1

Reviewer 1 Report

There are few observations in the manuscript.

Review some observations (ADMET, SEM) that are not defined.

Include a brief explanation of the purpose and selection of liquid additive (MeCN and DMSO).

Reviewer 2 Report

The study by Canale et al. deals with the mechanochemical synthesis, in vitro pharmacological properties, and glioprotective activity of a new small series (ten final compounds) of 1-(arylsulfonyl-isoindol-2-yl)piperazines targeting the 5-HT6R, which possess an interesting functional profile (biased ligand). Of note, the three most potent compounds (i.e., 3e, 3f, and 3g) showed a selective binding profile towards selected off-targets and suitable preliminary metabolic stability and safety profile. Finally, the most interesting compound 3g was selected to test its glioprotective properties against 6-OHDA-and ROT-induced toxicity in C8-D1A astrocyte cell line.

The manuscript is well-written and organized. Indeed, the brief introduction is informative, the rational design of the newly synthesized ligands is neat, and the experimental part is clear and well supported by the supplementary material, while results and discussion are appropriately interpreted. My only criticism is about the lack of molecular modeling studies to support the speculation that the authors made on compounds’ binding modes. Nevertheless, in my opinion, the current study will be of relevance for the journal’s readers and hence deserves to be published.

My suggestions are as follows:

1.     The Authors speculate on the possibility of a halogen bond or van der Waals interaction that might help to stabilize the ligand-receptor complex, however, they did not provide any experimental evidence about their statement. Also, they neither mentioned nor described any possible amino acid residue within the binding pocket involved in the proposed ligand-protein binding modalities. I can assume that their hypothesis relies on the previous results concerning a different set of ligands. Therefore, to further improve the quality of the manuscript, I may suggest providing binding pose analysis to corroborate SARs (i.e., lines 144-160, and lines 219-223). Particularly, since the Authors outlined the influence of the inversion and incorporation of sulfonamide into isoindoline scaffold (the current adopted strategy in this new series) as a stractural determinnat to  swicht the functional activity of the new compounds (from inverse agonist to neutral antagonist), SB-258585 should be included in the docking study as a reference compound.

2.   Table 3, line 271. Why did not Authors provide the exact value of IC50 (cytotoxicity) for the tested compounds as for the reference drug doxorubicin?

3.    Section 2.6, lines 294-296. It is not clear why did authors select the specific “non-toxic” dose of 0.25 μM for the glioprotective properties testing. Please, specify it accordingly.

4.     Line 29: MTT acronym should be explicated previously in line 282 where it is first mentioned.

5.  A few typos are present though the text, please carefully proofread the manuscript. As an example,  line 30: space missing (i.e., “the5-HT6R operated”); similarly “of5-HT6R bias” (line 63).
